# Dual Function of Secreted APE1/Ref-1 in TNBC Tumorigenesis: An Apoptotic Initiator and a Regulator of Chronic Inflammatory Signaling

**DOI:** 10.3390/ijms23169021

**Published:** 2022-08-12

**Authors:** Sunga Choi, Yu-Ran Lee, Ki-Mo Kim, Euna Choi, Byeong-Hwa Jeon

**Affiliations:** 1Department of Bioinformatics and Biosystems, Seongnam Campus of Korea Polytechnics, Seongnam-si 13122, Korea; 2Research Institute of Medical Sciences, College of Medicine, Chungnam National University, Daejeon 35015, Korea; 3Department of Medical Science, College of Medicine, Chungnam National University, Daejeon 35015, Korea; 4Korean Medicine Convergence Research Division, Korea Institute of Oriental Medicine (KIOM), Daejeon 34054, Korea; 5Department of Biology, Union University, Jackson, TN 38305, USA

**Keywords:** TNBC, PPTLS-APE1/Ref-1, apoptosis, tumor microenvironment, cytokines, anti-inflammatory signaling

## Abstract

The simultaneous regulation of cancer cells and inflammatory immune cells in the tumor microenvironment (TME) can be an effective strategy in treating aggressive breast cancer types, such as triple-negative breast cancer (TNBC). Apurinic/apyrimidinic endonuclease 1/redox effector factor 1 (APE1/Ref-1) is a multi-functional nuclear protein that can be stimulated and then secreted. The extracellular APE1/Ref-1 causes a reduction in disulfide bonds in cytokine receptors, resulting in their conformational changes, thereby inhibiting inflammatory signaling. Furthermore, the secreted APE1/Ref-1 in response to acetylation has been shown to bind to a receptor for the advanced glycation end product (RAGE), initiating the apoptotic cell death of TNBC in vitro and in vivo. This study used PPTLS-APE1/Ref-1 in an adenovirus vector (Ad-PPTLS-APE1/Ref-1) for the constant expression of extracellular APE1/Ref-1, and our results demonstrated its dual function as an apoptotic initiator and inflammation regulator. Injecting MDA-MB 231 orthotopic xenografts with the Ad-PPTLS-APE1/Ref-1 inhibited tumor growth and development in response to acetylation. Moreover, Ad-PPTLS-APE1/Ref-1 generated reactive oxygen species (ROS), and tumor tissues derived from these xenografts exhibited apoptotic bodies. Compared to normal mice, a comparable ratio of anti- and pro-inflammatory cytokines was observed in the plasma of Ad-PPTLS-APE1/Ref-1-injected mice. Mechanistically, the disturbed cytokine receptor by reducing activity of PPTLS-APE1/Ref-1 inhibited inflammatory signaling leading to the inactivation of the p21-activated kinase 1-mediated signal transducer and activator of transcription 3/nuclear factor-κB axis in tumor tissues. These results suggest that the regulation of inflammatory signaling with adenoviral-mediated PPTLS-APE1/Ref-1 in tumors modulates the secretion of pro-inflammatory cytokines in TME, thereby inhibiting aggressive cancer cell progression, and could be considered as a promising and safe therapeutic strategy for treating TNBCs.

## 1. Introduction

A basal-like subtype of breast cancer, triple-negative breast cancer (TNBC), comprises approximately 20% of all breast cancers worldwide [1]. TNBC is aggressive and is often associated with a high proliferation, a high histological grade, and a poor prognosis. The lack of targetable receptors, such as estrogen, progesterone, and epidermal growth factors [2], makes chemotherapy the only available systemic treatment for TNBC. The cells respond to chemotherapeutic treatment in early-stage TNBC; however, they become rapidly resistant, posing a significant challenge.

Chronic inflammation is a hallmark of cancer and can be destructive, promoting the malignant transformation of the tumor [3]. The aggressiveness of TNBC is gained by the surrounding tumor microenvironment (TME); several pro-inflammatory mediators, including tumor necrosis factor (TNFs), interferons (IFN), interleukins (IL), and chemokines (CXC/CCL) have been shown to participate in tumorigenesis, even though their pleiotropic roles in biological processes remain unclear [4]. The TNF-α and TNF-α receptor (TNFR) complex induces a range of inflammatory genes by activating the nuclear factor-κB (NF-κB) pathway in inflammation-induced tumor growth and metastasis [5,6]. IL-6 and IL-8 lymphokines derived from T cells regulate the proliferation and neovascularization in ovarian carcinoma SKOV-3 and TNBC cell lines [7,8,9]. Moreover, elevated IL-6 levels have been reported to be evident in a patient with metastatic breast cancer in the advanced stage [10]. A marked increase in the transforming growth factor (TGF)-β production has been correlated with an increased invasiveness in patients with breast cancer [11].

Since many cytokines, cytokine-producing cells, and immune cells play critical roles in tumor development, increasing attention has been paid to modulating cytokine levels to combat tumorigenesis. Immunological modulation of cytokine-mediated inflammation in malignant breast tumors showed tumor immune evasion by escaping T-cell effector function. Combined inhibition of IL-6 and IL-8 was reported to be effective for treating TNBC in vivo [12]. Direct targeting of the TNF-α gene in Hs578T breast cancer cells increased the levels of pro-apoptotic factors, demonstrating that the modulation of various cytokines could induce tumor cell death [13]. Inhibiting the epidermal growth factor receptor (EGFR) and vascular endothelial growth factor (VEGF)-A signaling using specific antibodies also improved the anti-tumor effect in TNBC [14,15]. 

Apurinic/apyrimidinic endonuclease 1/redox effector factor 1 (APE1/Ref-1) is a nuclear protein that plays a vital role in DNA repair and redox regulation against oxidative stress [16,17]. It has been demonstrated that the APE1/Ref-1 protein could be translocated into the cytoplasm and then secreted extracellularly [18,19,20,21]. Moreover, we have reported that auto- and paracrine APE1/Ref-1 secreted by TNF-α-stimulated endothelial cells could disrupt the inflammatory TNFR1 signaling by reducing their disulfide bonds in the extracellular domain [22]. Similarly, APE1/Ref-1 secreted by lipopolysaccharide (LPS)-stimulated macrophages led to thiol/disulfide bond exchange in IL-1 receptors or Toll-like receptors functioning as reducing agents and effectively attenuating inflammatory responses [22]. These results were corroborated by our recent data that used a reduced form of recombinant APE1/Ref-1, which showed the inhibition of TNF-α signaling in the TNBC microenvironment [23]. In addition, we have demonstrated that extracellular APE1/Ref-1 binds to its receptor for advanced glycation end products (RAGE) in response to acetylation in vitro [19] and in vivo [20] models. Our results suggest the role of extracellular APE1/Ref-1 as an initiator of apoptosis in TNBC cells. More recently, we designed an adenovirus encoding PPTLS-APE1/Ref-1, which can be actively secreted extracellularly. Anti-inflammatory effects of the secreted PPTLS-APE1/Ref-1 were proved with a decreased level of pathogen-caused inflammation in an LPS-stimulated sepsis model [24].

In this study, we hypothesized that secreted PPTLS-APE1/Ref-1 could regulate the TME of breast cancer in vivo and play a critical role in initiating apoptosis in response to acetylation. We determined the plasma level of inflammation-associated cytokines in MDA-MB-231 orthotopic xenografts and evaluated tumor growth retardation and cell death following the intravenous administration of adenovirus encoding PPTLS-APE1/Ref-1.

## 2. Results

### 2.1. Adenovirus-Mediated PPTLS-APE1/Ref-1 Bound to RAGE in Response to Acetylation-Caused Apoptotic Cell Death

Our previous study showed that the acetylation of TNBC cells results in apoptotic cell death initiated by secreted and acetylated APE1/Ref-1 molecules [19,20]. In this study, we used adenovirus-mediated actively secreted PPTLS-APE1/Ref-1 molecules and determined their effects on the viability and DNA fragmentation of TNBC cells. The viability of MDA-MB-231 cells infected with Ad-PPTLS-APE1/Ref-1 (extracellular form) and treated with acetylsalicylic acid (ASA, acetyl group donor) showed a ~36% decrease compared with those infected with either Ad-β-galactosidase or Ad-APE1/Ref-1 (mainly intracellular form) (Figure 1A). Interestingly, MDA-MB-231 cells infected with Ad-APE1/Ref-1 but not treated with ASA showed a remarkable increase in viability compared with the Ad-β-galactosidase control cells. This result is consistent with the known function of intracellular APE1/Ref-1 as a repair and redox protein in various tumors [25,26].

The effects of Ad-PPTLS-APE1/Ref-1 on MDA-MB-231 cell death were confirmed by cytoplasmic histone-associated DNA fragmentation (Figure 1B). DNA fragmentation was intensified by more than 2-fold in cells exposed to ASA compared with the control cells. ASA treatment also increased apoptotic DNA fragmentation in Ad-β-galactosidase and Ad-APE1/Ref-1-infected cells. The results imply that acetylation induced the secretion of endogenous and exogenous APE1/Ref-1 in the control cells, resulting in increased apoptosis. Altogether, these data indicate the role of actively secreted PPTLS-APE1/Ref-1 and intracellular APE1/Ref-1 in accelerating the apoptotic death of TNBC cells when acetylated. The changes in cell viability and DNA fragmentation were observed in other TNBC cell lines, MDA-MB-468 and BT 549. When treated with ASA, those cells encoding PPTLS-APE1/Ref-1 showed viabilities of 61.7% and 32.5%, respectively (Figure 1A). These values were lower than that of adenovirus-infected cells without ASA treatment, albeit the difference was insignificant (*p* = 0.53). However, both cell lines secreting PPTLS-APE1/Ref-1 showed a ~2-fold increase in the number of apoptotic cells compared with those injected with β-galactosidase or APE1/Ref-1 (Figure 1B). Among the three cell lines, BT 549 showed greater differences in the cell viability of control cells (Ad-βgal and Ad-APE1/Ref-1) after acetylation than Ad-PPTLS-APE1/Ref-1-infected cells. This result could be explained by our previous finding proving that BT 549 induces the upregulation of RAGE in the plasma membrane in response to acetylation and enhances the secretion of endo- or exogenous APE1/Ref-1 molecules [19,20].

TNBC cells expressing PPTLS-APE1/Ref-1 showed significantly increased apoptotic cell death after treatment with ASA; therefore, we determined whether this cell death was mediated by the binding of PPTLS-APE1/Ref-1 to RAGE, a transmembrane receptor, by immunoprecipitation. Binding of secreted APE1/Ref-1 with RAGE was detected in all cells; however, their interaction was significantly elevated in PPTLS-APE1/Ref-1-infected cells following acetylation (Figure 1C). This is likely due to the fact that the extracellular form of APE1/Ref-1 is present at a much higher level in PPTLS-APE1/Ref-1-infected cells than in control cells, where intracellular APE1/Ref-1 underwent acetylation and secretion [24]. These results indicate that the interaction between PPTLS-APE1/Ref-1 and RAGE is responsible for the increased apoptotic cell death in the infected cells.

### 2.2. Correlation between PPTLS-APE1/Ref-1 Levels in Blood and the Inhibition of Tumor Growth

We next investigated the role of adenovirus-mediated PPTLS-APE1/Ref-1 as an initiator of apoptotic signaling in vivo. We first determined whether PPTLS-APE1/Ref-1 could be detected in the blood of MDA-MB-231 xenografts and whether these mice had inhibited tumor growth. MDA-MB-231 xenografts received ASA oral gavage (three times/week) and an intravenous injection of Ad-PPTLS-APE1/Ref-1 (two times/week) alternatively for 7 weeks (Figure 2A). Necrotic cell death was prevented by choosing an optimal concentration of ASA or adenovirus titers were cautiously determined as previously described [20,24]. Tumor development was monitored using whole-body IVIS post-tumor-cell implantation. The average body weights of salicylic acid (SA, lacking acetyl group), ASA-treated, and adenovirus-injected mice did not significantly differ throughout the experimental period with no noticeable physical signs (Figure 2B). The average body weights of the PPTLS-APE1/Ref-1-injected mice without tumors were higher than those of SA, ASA-treated, or Ad-β-galactosidase-injected mice, albeit the values were not statistically significant (Figure 2C). These data indicate the normal growth of mice administered with adenovirus encoding PPTLS-APE1/Ref-1, APE1/Ref-1, or β-galactosidase. Notably, the average wet tumor weight of Ad-PPTLS-APE1/Ref-1-injected mice (0.62 g) was significantly lower than that of Ad-β-galactosidase-injected mice (1.58 g, *p* < 0.05; Figure 2D). 

At the experimental endpoint, the level of PPTLS-APE1/Ref-1 was measured from xenograft plasma (Figure 2E). The levels of APE1/Ref-1 in the plasma were ~1.8-fold higher in Ad-PPTLS-APE1/Ref-1-injected xenografts (3.4 ng/100 μL) than in Ad-APE1/Ref-1-injected xenografts (1.8 ng/100 μL), which had similar levels to that in ASA-treated xenografts (1.9 ng/100 μL) or Ad-β-galactosidase-injected xenografts (1.7 ng/100 μL). These results demonstrate that PPTLS-APE1/Ref-1 was actively secreted from PPTLS-APE1/Ref-1 xenografts, leading to the retardation of tumor cell growth by triggering apoptosis in vivo.

### 2.3. Retarded Tumor Cell Growth in Xenografts Secreting PPTLS-APE1/Ref-1 into the Blood

The real-time effects of actively secreted PPTLS-APE1/Ref-1 molecules on the tumors of MDA-MB-231 orthotopic xenografts were recorded using IVIS after luminescent luciferin injection. The relative luminescence intensities correlated with the size of tumor growth. No mouse died from any unexpected disease during the 12-week study period. Tumor development was evident 7 days post-implantation of luminescent MDA-MB-231 cells (Figure 3A). Tumor growth significantly decreased in Ad-PPTLS-APE1/Ref-1-injected mice compared with ASA-treated, Ad-β-galactosidase, or Ad-APE1/Ref-1-injected mice (Figure 3A,B). The significant decrease in the tumor weight of Ad-PPTLS-APE1/Ref-1-injected mice at the endpoint (Figure 2D) was associated with a negligible level of luminescent tumor cells at weeks 6–12 and differed dramatically compared with the Ad-β-galactosidase-injected control groups (Figure 3A,B). In contrast, a remarkable increase in luminescent tumor cells was observed in ASA-treated or Ad-APE1/Ref-1-injected mice over time. A whole-body metastasis accompanied the uncontrolled growth of tumor cells in a few SA-treated or Ad-β-galactosidase-injected mice, causing the death of the mice at weeks 7–8 following injections (Figure 3A,B).

Next, the induction of apoptotic cell death via reactive oxygen species (ROS) generation was investigated by terminal deoxyribonucleotide transferase-meditated dUTP nick end labeling (TUNEL) and dihydroethidium (DHE) staining in the mice groups. The histological findings showed that Ad-PPTLS-APE1/Ref-1-injected mice had a ~1.5-fold higher count of apoptotic bodies than ASA-treated mice or Ad-APE1/Ref-1-injected mice (Figure 3C,D). Previous studies have demonstrated that intracellular signaling following RAGE stimulation via ligand binding led to ROS generation [19,20]; therefore, we assessed the intensity of DHE staining of fresh frozen tumor sections. More red intense staining was observed in Ad-PPTLS-APE1/Ref-1-injected tumor tissues than in SA-treated mice or Ad-β-galactosidase-injected mice, indicating that enhanced ROS generation in these mice caused apoptotic cell death. In contrast, Ad-APE1/Ref-1-injected mice with less intense TUNEL staining were not ROS-positive. Noticeable apoptotic bodies or DHE-positive red staining were not observed in tumor tissues from SA-treated mice or Ad-β-galactosidase-injected mice (Figure 3C,D). Collectively, these data demonstrated that secreted PPTLS-APE1/Ref-1 molecules suppress tumor growth and spontaneous metastases in MDA-MB-231 xenografts in vivo and induce increased apoptosis by ROS generation. These results suggest that the therapeutic intervention of Ad-PPTLS-APE1/Ref-1 could be beneficial in treating patients with breast cancer.

### 2.4. Plasma PPTLS-APE1/Ref-1 in Cytokine Regulation

PPTLS-APE1/Ref-1 exhibited redox activity that modified various cytokine receptors in the blood of LPS-stimulated septic mice [24]. To investigate the effect of PPTLS-APE1/Ref-1 in modulating cytokine production in TME, we determined the levels of the following inflammatory-tumor-associated cytokines: IFN-γ, IL-1β, IL-6, IL-12, TNF-α, MCP-1/CCL2, MIP-2/CXCL2/CINC-3, CXCL12, IL-2, and IL-10. The Ad-PPTLS-APE1/Ref-1-injected mice showed decreased levels of IL-1β and IL-6, pro-inflammatory cytokines, compared with SA-treated or Ad-β-galactosidase-injected mice (Figure 4). The plasma IL-6 level of Ad-PPTLS-APE1/Ref-1-injected mice was ~3.3-fold lower (10 pg/mL) than that of SA-treated or Ad-β-galactosidase-injected mice (32 and 33 pg/mL, respectively). However, one of the Ad-APE1/Ref-1 xenografts had a decreased level of IL-6. These results strongly support the role of secreted PPTLS-APE1/Ref-1 in regulating inflammatory signaling in the blood and its role in reducing the disulfide bonds of the extracellular domain of inflammatory cytokine receptors, such as the IL-6 receptor.

Interestingly, the anti-inflammatory cytokines IL-2 and IL-10 levels significantly increased in Ad-PPTLS-APE1/Ref-1-injected mice. Even though not significantly, Ad-APE1/Ref-1-injected mice also showed increased levels of IL-2 and IL-10. These results suggest that an autocrine and paracrine loop of secretory PPTLS-APE1/Ref-1 molecules is involved in regulating cytokines in the TME of MDA-MB-231 xenografts.

Since chemokines, such as CCL2 and CXCL12, can regulate tumor behavior similar to metastasis, their levels were also determined. Consistent with the results of pro-inflammatory cytokines, the levels of CCL2, which mediates macrophage-related inflammation, were significantly decreased. In contrast, CXCL12 was increased in Ad-PPTLS-APE1/Ref-1-injected mice compared with SA-treated or Ad-β-galactosidase-injected mice. These data support the role of PPTLS-APE1/Ref-1 in the downregulation of the pro-inflammatory signaling pathway in the TME.

### 2.5. Anti-Inflammatory Signaling in Tumors Expressing PPTLS-APE1/Ref-1 Derived from MDA-MB-231 Orthotopic Xenografts 

Since the production of inflammatory cytokines in the plasma was downregulated in the TME, we investigated inflammatory signaling in tumors derived from Ad-PPTLS-APE1/Ref-1-injected xenografts. STAT3 and NF-κB are molecular surrogates of downstream inflammatory signaling and function as inflammatory regulators. PAK1 protein, a family of serine/threonine p21-activated kinases, is a regulator of cancer-inflammation signaling leading to the activation of STAT3 and NF-κB [27]. The levels of phosphorylated STAT3 (p-STAT3) and NF-κB (p65) dramatically decreased in the tumors of Ad-PPTL-SAPE1/Ref-1-injected xenografts (Figure 5A). The expression of phospho-PAK1, phospho-STAT3, and p65 NF-κB were upregulated in the tumors of SA-treated or Ad-β-galactosidase-injected mice. However, their levels in the tumors of Ad-PPTLS-APE1/Ref-1-injected mice were significantly decreased showing less than 50% of those observed in Ad-β-galactosidase-injected mice (46%, 55%, and 37.5%, respectively) compared to Ad-β-galactosidase-injected mice (Figure 5B). The cancer-inflammatory signaling was also downregulated in Ad-APE1/Ref-1-injected mice, showing comparable levels with that in Ad-PPTLS-APE1/Ref-1-injected mice. These results indicate that the pharmacological targeting of PPTLS-APE1/Ref-1 may provide an effective strategy for regulating the cancer inflammatory PAK1–STAT3-NF-κB axis in vivo.

## 3. Discussion

This study has provided experimental and preclinical evidence demonstrating the importance of the adenovirus-mediated gene delivery and the secretion of PPTLS-APE1/Ref-1 to induce apoptotic cell death in TNBC in vivo. It is speculated that the PPTLS-APE/Ref-1 reduces the disulfide bonds of the ligand-binding domain of cytokine receptors and blocks the PAK1–STAT3/NF-κB axis, thereby suppressing inflammatory signaling in TME.

Adenovirus-based gene therapy is considered to have high potential efficacy for treating various cancers including breast cancer [28,29]. However, the results of the in vivo trials were disappointing due to their low efficiency, non-specific transgene integration, and decreased replicability of the viral vector [30]. 

In a previous study, we designed and employed a novel concept where an adenovirus encoding PPTLS-APE1/Ref-1 with a preprotrypsin leader sequence was produced as an actively secreted protein without a stimulator [24]. Infecting normal and tumor cells with an adenovirus encoding PPTLS-APE1/Ref-1 confirmed their efficient expression, secretion, and systemic circulation of the protein. In contrast, APE1/Ref-1 was over-expressed within cells, secreted extracellularly in response to acetylation, bound to RAGE, and caused intracellular signaling as shown in vitro [19] and in vivo [20]. However, actively secreted PPTLS-APE1/Ref-1 acted similar to acetylated APE1/Ref-1. No cytotoxicity from the adenovirus injection was observed by analyzing various cytokine levels compared to normal mice [24]. Collectively, these results suggest that the Ad-PPTLS-APE1/Ref-1 can be periodically applied to affected individuals with proper treatment schedules. 

In our previous reports, extracellular functions of APE1/Ref-1 were demonstrated; as a reductant causing thiol/disulfide bond exchange in the absence of an acetyl group donor or an apoptotic initiator in the presence of an acetyl group donor. Consequently, we designed and infected normal cells to express PPTLS-APE1/Ref-1 instead of targeting cancer cells for specific APE1/Ref-1 expression. With ASA treatment, acetyl moiety was quickly absorbed and distributed to cells within one hour, resulting in the post-translational modification of cellular proteins but the modification quickly returned to the baseline, losing an acetyl group [31]. To take advantage of the dual functions of APE1/Ref-1 determined by its acetylation status, we performed an alternate administration of adenovirus injection and ASA. In this therapeutic model, either one of the events was expected to occur: ASA administration after adenovirus injection produces Ac-PPTLS-APE1/Ref-1 which binds to RAGE and induces apoptotic signaling or with adenovirus injection alone, newly secreted PPTLS-APE1/Ref-1 in the blood without acetyl group interferes with inflammation-related cytokine receptors with reducing power. Our previous finding supports the role of PPTLS-APE1/Ref-1 as a reducing molecule, where the oxidized form of secreted PPTLS-APE1/Ref-1(C65A/C93A) redox mutant showed no anti-inflammatory activity in LPS-induced myeloperoxidase activity [21,24]. At the endpoint of TNBC xenografts treatment, differences in the tumor growth and the levels of immune cytokine existed between different mice groups, suggesting the dual function of extracellular PPTLS-APE1/Ref-1 depending on acetylation.

Increased PAK expression has been demonstrated in invasive tumor tissues including TNBC, implicating the effects of PAK activity on cell growth, survival, and motility [32,33,34]. Notably, PAK1 is also an upstream effector of STAT3 and NF-κB, associated with inflammation-associated malignancy, suggesting its role in inflammation-related tumor progression [35,36]. The regulation of PAK1 using small compound inhibitors, shRNA, or miRNA not only suppressed tumor growth but also upregulated the tumor immune response. Our data demonstrated that PPTLS-APE1/Ref-1 regulated the inflammatory signaling through cytokine receptors by suppressing phosphorylated PAK1–STAT3/p65 NF-κB as shown in tumor tissues from TNBC xenografts. These results clearly show that PPTLS-APE1/Ref-1 can inhibit pro-inflammatory signaling and alter the secretion of cytokines and chemokines into the TME, consequently preventing the action of pro-inflammatory cytokines.

In summary, this study addressed how the secreted adenovirus-mediated protein correlated with preclinical outcomes in vivo, establishing the role of PPTLS-APE1/Ref-1 as a reducing agent and a cell death signaling initiator of TNBC. In addition, PPTLS-APE1/Ref-1 remarkably inhibited inflammatory signaling in tumor tissues by suppressing PAK1–STAT3/NF-κB signaling, accompanied by the downregulation of inflammatory cytokines in the TME. These results can provide a basis for applying APE1/Ref-1 to protein-mediated pharmaceutical therapy. Our study strongly highlights the therapeutic potential of the APE1/Ref-1 protein agonist as an adjuvant/neoadjuvant in treating patients with TNBC. The knowledge gained in this study will shed light on further expanding its clinical therapeutic applications using protein pharmaceuticals.

## 4. Materials and Methods

### 4.1. Cell Culture

Human breast adenocarcinoma cell lines (MDA-MB-231, MDA-MB-468, and BT-549) and luciferase-expressing MDA-MB-231 cells (MDA-MB-231-Red-FLuc Bioware^®^ Brite) were obtained from ATCC (Manassas, VA, USA) and PerkinElmer (Waltham, MA, USA), respectively. All cell lines at a passage number less than 20 were authenticated by short tandem-repeat profiling and cell morphology assessment. The MDA-MB-231 and MDA-MB231-FLuc cells were cultured in RPMI 1640 medium (Gibco) supplemented with 10% fetal bovine serum (FBS). The BT-549 and MDA-MB-468 cells were maintained in DMEM (Gibco, Grand Island, NY, USA) containing 0.01 mg/mL human recombinant insulin and L-15 medium (Gibco) with 10% FBS, respectively.

### 4.2. Cell Viability and Apoptotic DNA Fragmentation Assay 

Adenovirus expressing β-galactosidase, APE1/Ref-1, or PPTLS-APE1/Ref-1 was prepared as previously described [24]. The effect of each adenovirus-mediated protein expression on cell viability and DNA fragmentation of MDA-MB-231, MDA-MB-468, and BT-549 cells was determined using a RealTime-Glo MT luminescent kit (Promega, Madison, WI, USA) and Cell Death Detection ELISA kit (Roche Applied Science, Indianapolis, IN, USA), respectively. 

APE1/Ref-1 derived from cells expressing an adenoviral-mediated protein in the culture supernatant or whole-cell lysates was immunoprecipitated using anti-RAGE (Cell Signaling Technology, Beverly, MA, USA), followed by immunoblotting using anti-APE1/Ref-1 or anti-RAGE antibodies as described previously [19]. 

### 4.3. Animal Experiments

Female BALB/c nude mice (eight-week-old, 16–18 g) were purchased from Chung-Ang Laboratory Animal (Seoul, Korea). Animals were housed at 24 °C with a 12-h day/night cycle under specific pathogen-free conditions and had *ad libitum* access to a γ-ray-irradiated laboratory rodent diet (Purina Korea) and autoclaved water. All experiments were performed following the relevant guidelines and regulations of the animal care unit at Chungnam National University. All animal protocols were approved by the Ethics Committee of Animal Experimentation of Chungnam National University (approval number: CNUH-016-A0015).

The MDA-MB-231-Red-Fluc TNBC cell line was used to generate orthotropic xenografts. On day zero, TNBC cells (1.5 × 10^6^ cells/mouse; 1:1 ratio with Matrigel) were unilaterally injected into the mammary fat pad as previously described [20]. Three days after injection, tumor implantation was verified by IVIS imaging (Lumina XRMS instrument, PerkinElmer), and groups were stratified (4 mice/group). The experimental schedule is described in Figure 2A. For treating tumor mouse models, three adenoviruses were prepared: Ad β-galactosidase as a virus infection control, Ad-APE1/Ref-1 mainly as an intracellular form, and Ad-PPTLS-APE1/Ref-1 as an extracellular form that contains secretory signal sequence. The PPTLS-APE1/Ref-1 was constantly secreted for the desired period [24]. After the confirmation of the tumor, 50 μL of Ad β-galactosidase, Ad-APE1/Ref-1 (2 × 10^9^ IFU), or Ad-PPTLS-APE1/Ref-1 (2 × 10^9^ IFU) in PBS was intravenously injected twice per week. On the following day of Ad virus injection, the mice were orally administered 20 mg SA or ASA/(kg/day) three times per week. For long-term real-time detection of tumors, bioluminescent images were taken weekly with an integration time of 5 sec. Images were quantified and normalized using vendor software (Living Image^®^ 4.0, Caliper Life Sciences, Waltham, MA, USA). A luminescent camera was set to obtain images every 30 s with medium binning and 1 full stop, and a blocked excitation and open emission filter. Region-of-interest (ROI) of the same size and shape were used for all mice throughout the study. The bioluminescence images were quantified by measuring the total photons over the prostate region, and the average photon flux within the ROI was presented as photons/sec/cm^2^/steradian. Mouse weight and tumor dimensions were recorded periodically. At the experimental endpoint, tumor tissues were harvested and used for histological and immunoblot analyses. The animal experiments were repeated twice with similar results.

### 4.4. Measurement of APE1/Ref-1 and Cytokine Levels in Xenograft Plasma

Cell-free plasma was obtained from the blood of each xenograft tumor by centrifugation at 4000× *g* at 20 °C. The level of APE1/Ref-1 in each sample was quantified by ELISA as previously described [20]. Each sample was assayed in duplicate, and the mean values were determined. For simultaneous measurement of soluble cytokines in the plasma, a multiplex bead assay was performed using MultiPlex kits (KOMA Biotech, Seoul, Korea).

### 4.5. Histological Analysis

To observe apoptotic cell death in tissues, tumors from xenografts were embedded in paraffin and sectioned at 5 μm. The presence of apoptotic bodies within the tumor sections was determined by TUNEL assay using an ApopTag Plus peroxidase in situ apoptosis kit (EMD Millipore, Billerica, MA, USA). Alternatively, fresh tumor samples were fixed in an acetone solution. Frozen tissues were cut into 30 μm sections and treated with dihydroethidium (DHE; Thermo Fisher Scientific) for 7 min at 37 °C in the dark. Images were obtained immediately after mounting. The red DHE fluorescence intensity of at least 100 blue DAPI nuclei per sample was scored using Image J in at least two mice per experimental condition.

### 4.6. Immunoblotting

Tumor tissues harvested from the control (SA-treated) or adenovirus-injected mice combined with ASA treatment were suspended in PBS and homogenized using a sonicator (Hielscher, Teltow, Germany). Immunoblotting was performed as described previously [19]. Antibodies against phospho-PAK-1, phospho-STAT-3, PAK-1, and STAT-3 were purchased from Santa Cruz Biotechnology Inc (Santa Cruz, CA, USA). Anti-β-actin (AC-74) antibodies were from Sigma-Aldrich. Antibodies against phospho-NFkB (Cat. no. 9662) and acetyl-lysine (Cat. no. 9441) were from Cell Signaling Technology (Danvers, MA, USA).

### 4.7. Statistical Analyses

Group means were compared using an unpaired *t*-test or one-way ANOVA, followed by Dunnett’s or Bonferroni’s multiple-comparison test. Statistical analyses were performed in GraphPad Prism version 5.01 (GraphPad Software, Inc., San Diego, CA, USA). Statistical significance was set at *p* < 0.05.

## Figures and Tables

**Figure 1 ijms-23-09021-f001:**
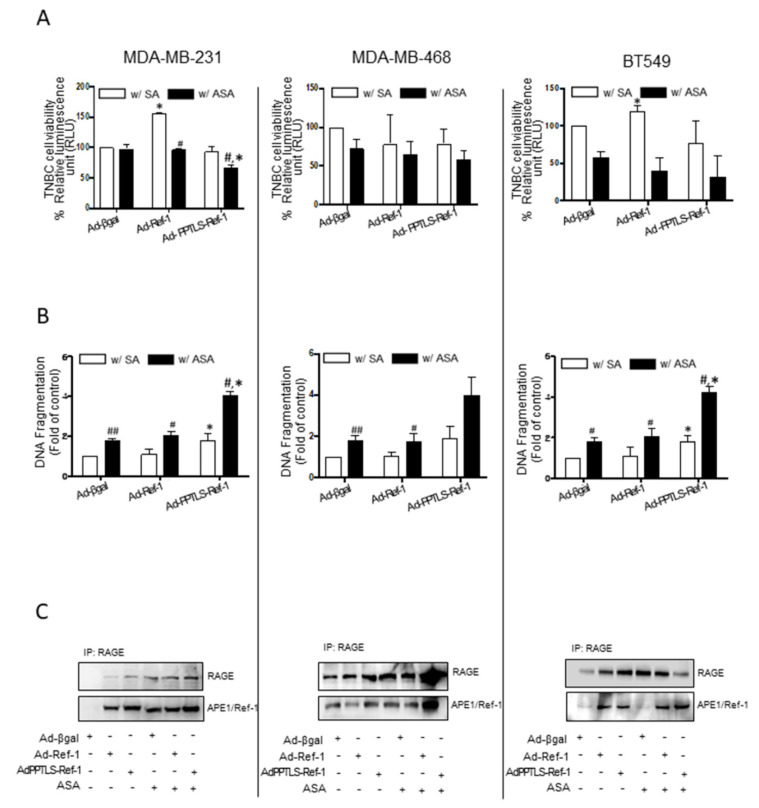
Adenovirus-mediated secretory PPTLS-APE1/Ref-1 affected the viability of TNBC cells in response to acetylation. TNBC cells encoding Ad-β-galactosidase (Ad-β-gal in the figures), Ad-APE1/Ref-1 (Ad-Ref-1), or Ad-PPTLS-APE1/Ref-1 (Ad-PPTLS- Ref-1) were treated with SA or 1 mM ASA for 24 h. Left panel: MDA-MB-231 cells, middle panel: MDA-MB-468 cells, right panel; BT549 cells. (**A**) Cell viability was determined using RealTime-Glo MT luminescent kit. (**B**) Apoptotic cell death in the TNBC cells was assessed by the quantification of cytoplasmic histone-associated DNA fragmentation. (**C**) Binding of the secreted APE1/Ref-1 to RAGE in the membrane fraction was observed by immunoprecipitation with RAGE antibody and detected by anti-RAGE or APE1/Ref-1 antibodies. *Columns*, mean (*n* = 3); *bars*, Standard errors (SE) ^#^, *p* < 0.05, ^##^, *p* < 0.01, significantly different from SA-treated cells within a group by unpaired *t*-test. *, *p* < 0.05, significantly different between SA- and ASA-treated groups, by one-way ANOVA followed by Bonferroni’s multiple comparison test. Similar results were observed in replicate experiments.

**Figure 2 ijms-23-09021-f002:**
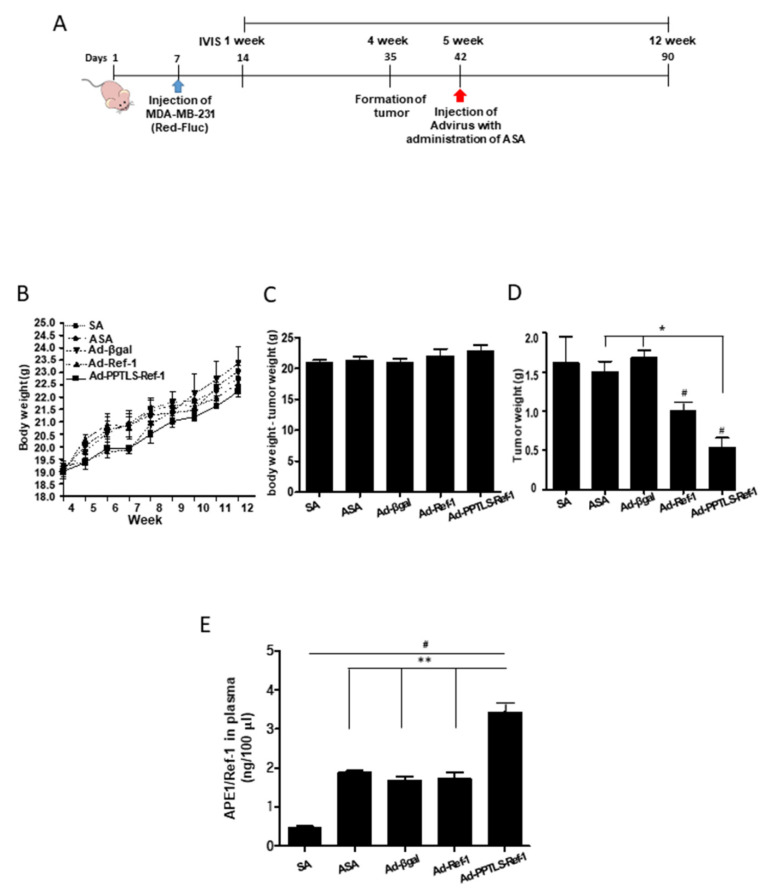
Xenografts treated with Ad-PPTLS-APE1/Ref-1 and ASA significantly inhibited tumor growth. (**A**) Schematic representation of the treatment regime for intravenous injection of adenovirus (Ad virus) in MDA-MB-231 cells xenografts. Female nude mice were injected with MDA-MB-231-Fluc cells. After the formation of detectable tumors according to IVIS imaging (7 days post-injection), mice were weighed and randomly assigned to SA, ASA, or Ad virus with ASA-treated groups. The adenovirus encoding β-galactosidase (Ad-β-gal), APE1/Ref-1 (Ad-Ref-1), or PPTLS-APE1/Ref-1 (Ad-PPTLS-Ref-1) was injected twice per week. The virus injection was followed by the administration of 20 mg/(kg/day) ASA on the following day. (**B**–**D**) Effect of intravenous injection of Ad virus accompanied by oral administration of 20 mg/(kg/day) ASA on total body weight (**B**), body weight in the absence of tumors (**C**), and wet tumor weight (**D**). Data reflect the mean of the independent replicate experiment (*n* = 5–6 mice/group, with tumor cells orthotopically injected into the mammary fat pad of each mouse). (**E**) Plasma levels of APE1/Ref-1 were measured by ELISA. *Columns* mean (*n* = 3); *bar*, SE. ^#^, *p* < 0.05, significantly different from only the SA-treated group by one-way ANOVA, followed by Dunnett’s test. * *p* < 0.05, ** *p* < 0.01 significantly different between indicated groups by one-way ANOVA followed by Bonferroni’s multiple comparison test. Similar results were observed in replicate experiments.

**Figure 3 ijms-23-09021-f003:**
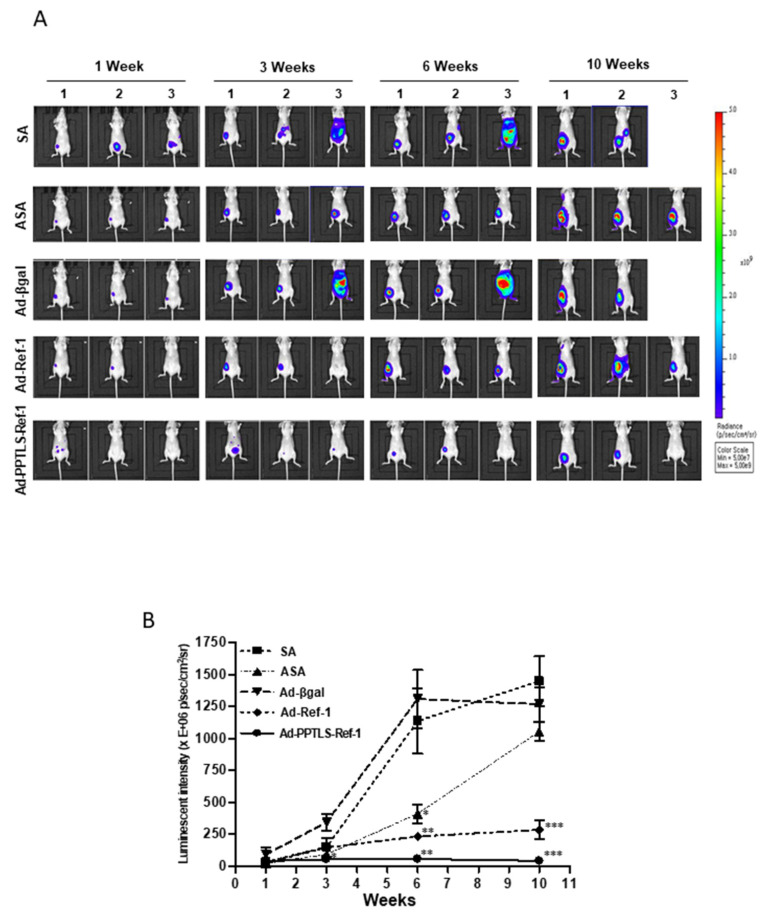
Xenografts treated with Ad-PPTLS-APE1/Ref-1 and ASA retarded tumor growth in vivo. The numbers on the top of the image represent individual xenografts. (**A**) Representative IVIS images obtained after luciferin injection at different days post-injection of MDA-MB-231-Fluc cells show reduced tumor growth in mice. (**B**) Tumor-growth curves for each group are shown for up to 10 weeks post-injection of MDA-MB-231-Fluc cells. Bioluminescence was expressed as flux (photons/s). The graph represents the mean values of 5–6 mice per group from two independent experiments; bars represent the SE. * *p* < 0.05, ** *p* < 0.01, *** *p* < 0.001, a significantly different from Ad-β-galactosidase with ASA-treated group by one-way ANOVA, followed by Bonferroni’s multiple comparison test. (**C**) Representative staining images of apoptotic bodies (TUNEL staining) and ROS generation (DHE staining, red fluorescence) in tumor sections from three mice in each group (magnification, ×400). (**D**) Quantitative analysis of apoptotic bodies and ROS generation using image J. The TUNEL-positive ratio was calculated by counting the positively stained cells and dividing them by the total cell number. *Columns*, mean (*n* = 3); *bars*, SE. #, *p* < 0.05, significantly different from only the SA-treated group by one-way ANOVA, followed by Dunnett’s test. * *p* < 0.05, significantly different between indicated groups by one-way ANOVA followed by Bonferroni’s multiple comparison test. Representative results from replicate experiments are shown.

**Figure 4 ijms-23-09021-f004:**
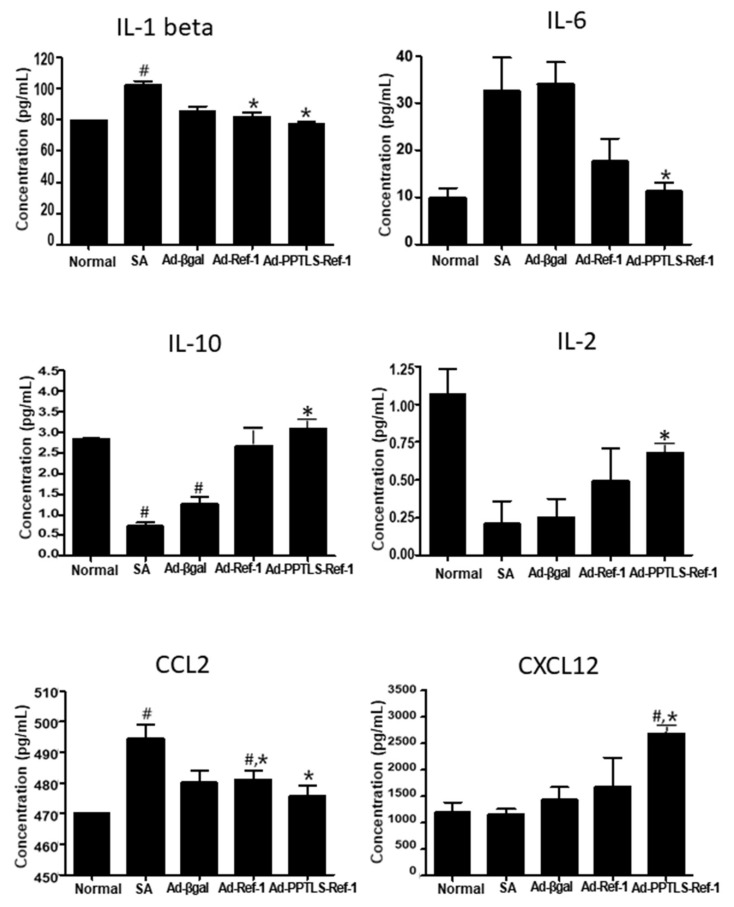
Multiplex cytokine profiling in xenograft plasma. Analysis of pro-inflammatory cytokines in TNBC xenografts. Plasma samples from xenografts were collected at the endpoint and analyzed. *Columns*, mean (*n* = 3–4); *bars*, SE. #, *p* < 0.05, significantly different from the normal group by one-way ANOVA followed by Dunnett’s test. *, *p* < 0.05, significantly different from only SA-treated or Ad-β-galactosidase with ASA-treated group by one-way ANOVA followed by Bonferroni’s multiple comparison test.

**Figure 5 ijms-23-09021-f005:**
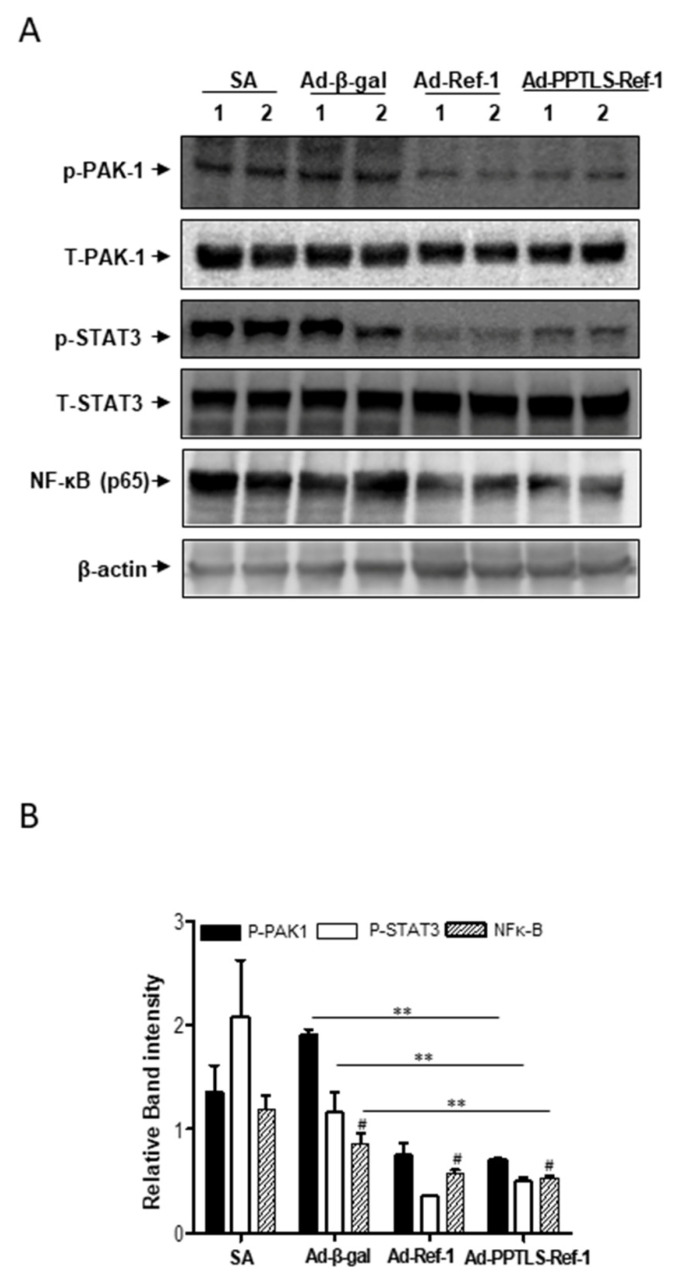
Apoptotic cell death in tumors from xenografts cascaded by the downregulation of PAK1. Immunological analysis of the tissues from MDA-MB-231 xenografts treated with Ad-β-galactosidase, Ad-APE1/Ref-1, or Ad-PPTLS-APE1/Ref-1and 20 mg/(kg/day) ASA, or SA alone. The numbers on the top of the band represent individual tumors from different xenografts. (**A**) Representative immunoblots showing PAK-1, STAT-3, and NF-κB p65 levels. β-actin was used as a protein-loading control. Immunoblotting was performed at least two times for each protein using independently prepared lysates with similar results. (**B**) The fold-changes in the levels of phosphorylated PAK1, STAT3, and NFκB p65 relative to the control are shown for each treatment. *Columns*, mean (*n* = 3); *bars*, SE. #, *p* < 0.05, significantly different from only the SA-treated group by one-way ANOVA followed by Dunnett’s test. **, *p* < 0.05, significantly different from Ad-β-galactosidase with ASA-treated groups, by one-way ANOVA followed by Bonferroni’s multiple comparison test.

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
