# Peer review of "Dual Function of Secreted APE1/Ref-1 in TNBC Tumorigenesis: An Apoptotic Initiator and a Regulator of Chronic Inflammatory Signaling"

_ijms, 2022, doi:10.3390/ijms23169021_

Round 1

Reviewer 1 Report

Following previous studies, this study established the role of the secreted adenovirus-mediated protein PPTLS-APE1/Ref-1 as an initiator of triple negative breast cancer cell death signaling and also showed that PPTLS-APE1/Ref-1 remarkably inhibits inflammatory signaling in tumor tissues through the suppression of PAK1–STAT3/NF-κB signaling, accompanied with down-regulation of inflammatory cytokines in the tumor microenvironment. There are some concerns as shown in the following:

(1) Some abbreviation words must give the full name first in the text (see below).

(2) Give the full name and a brief introduction of APE1/Ref-1 as in the previous paper.

(3) Give the Figure legends.

(4) Typos and others:

L22: APE1/Ref-1 reduces cytokines receptors -> reduce may cause misleading to decrease

L40: PPTLSAPE1/Ref-1-> PPTLS-APE1/Ref-1

*L102: ASA full name first

L118: with viabilities of 61.7% and 32.5%, respectively-> with viabilities of 61.7% and 32.5% for PPTLS-APE1/Ref-1 group, respectively

*L141: Correlation between PPTLS-APE1/Ref-1 levels in blood and inhibition of tumor growth-> bold letter

*L146-147: alternatively for nine weeks (Fig. 2A). -> 5 week -12 week =7 weeks

L150: SA full name first

*L184: (Fig. 3B) -> (Fig. 3A?)

*L196: not observed in SA-treated mice or Ad-β-galactosidase–? injected mice

*L234-235: Since the production of inflammatory cytokines in plasma was downregulated in the TME was inhibited?

**L243-244: their levels in the tumors of Ad-PPTLSAPE1/Ref-1-injected mice were significantly decreased to 46%, 55%, and 37.5%, respectively (Fig. 5B) -> not match with the data in Fig. 5B

L317: give the passage number of used cell lines

L348: 1.5 × 106 ; L353: 2×109

L386: +TSA full name

*L392: Antibodies against poly-ADP-ribose polymerase-1 393 (PARP-1; C-2-10) and Bax (6A7) were from BD Biosciences (Bedford, MA, USA). -> no data?

L412: References

*R4, R12, R15: 715-723, 3470-3480, 1571-1578. page number not consistent writing

*R23: no page number (Diseases. 2018 May 21;6(2):42. doi: 10.3390/diseases6020042.)

Author Response

The response file to the reviewer’s comments was attached.

Reviewer 2 Report

This is the continuation of a series of studies of the same authors on the role of secreted APE/Ref-1 in inflammation and cancer. In the current study the authors used the previously constructed adenovirus encoding PPTLS-APE1/Ref-1 (which can be actively secreted extracellularly) as a putative tool against triple-negative breast cancer.

The study is of interest, but there are some issues that need to be resolved before it can be considered for publication.

1. There are some grammar and syntax errors that need to be corrected. Most importantly, there are often some incomprehensible or unclear sentences that need to be rephrased in order for the reader to be able to follow the text (for examples, please refer to the uploaded file).

2. Figures are of poor quality. They are all need to be replaced by Figures of higher resolution.

3. The authors should provide legends to the Figures describing the data presented and especially giving information on the statistical analysis and the symbols appearing in the graphs.

4.  In Figure 3D, the bars do not seem to correspond to the fluorescence intensity in the respective DHE microscopic pictures of Figure 3C. Please clarify.

5. Ln243: decreased compared to what? The authors should be specific about which group is compared to which.

6. The authors should describe in details the role of each adenovirus used in the experimental setup: Why did they use an adenovirus expressing β-galactosidase and one expressing APE1/Ref-1 in parallel to the one expressing PPTLS-APE1/Ref-1? Which one served as the control, what is the difference between APE1/Ref-1 and PPTLS-APE1/Ref-1? The reader gets familiarized in the progress of reading, but it would be nice if the authors did not take this information for granted (by only citing reference 19); this information could be explicitly mentioned in the Materials and Methods or the beginning of the Results section.

7. In the same vein, the authors should define abbreviations such as SA or ASA and provide some details on their role to the reader. Citing their previous works in which this information is given is not sufficient.

Author Response

(The authors gave the same response as above.)

Round 2

Reviewer 2 Report

This is the revised version of a previously submitted manuscript. The authors have addressed most of my concerns raised during the previous round of the reviewing process.